# Maternal and newborn healthcare providers' work-related experiences during the COVID-19 pandemic, and their physical, psychological, and economic impacts: Findings from a global online survey

**Delphin Kolié**[1,2‡]*, **Aline Semaan**[3‡], **Louise-Tina Day**[4], **Thérèse Delvaux**[3], **Alexandre Delamou**[1,2], **Lenka Benova**[3]

**1** African Centre of Excellence for the Prevention and Control of Communicable Diseases, University of Conakry, Conakry, Guinea, **2** Ministry of Health, Centre National de Formation et de Recherche en Santé Rurale de Maferinyah, Forecariah, Guinea, **3** Department of Public Health, Institute of Tropical Medicine, Antwerp, Belgium, **4** Maternal and Newborn Health Group, Department of Infectious Disease Epidemiology, London School of Hygiene & Tropical Medicine, London, United Kingdom

‡ DK and AS share first authorship on this work.
* dkolie@maferinyah.org

**Data Availability Statement:** Due to ethical constraints, the data underlying this analysis cannot be made publicly available. The dataset

## Abstract

The COVID-19 pandemic continues to have substantial impacts on health systems globally. This study describes experiences during the COVID-19 pandemic, and physical, psychological and economic impacts among maternal and newborn healthcare providers. We conducted a global online cross-sectional survey of maternal and newborn healthcare providers. Data collected between July and December 2020 included demographic characteristics, work-related experiences, and physical, psychological, and economic impacts of COVID-19. Descriptive statistics of quantitative data and content analysis of qualitative data were conducted. Findings were disaggregated by country income-level. We analysed responses from 1,191 maternal and newborn healthcare providers from 77 countries: middle-income 66%, high-income 27%, and low-income 7%. Most common cadres were nurses (31%), midwives/nurse-midwives (25%), and obstetricians/gynaecologists (21%). Quantitative and qualitative findings showed that 28% of respondents reported decreased workplace staffing levels following changes in staff-rotation (53%) and staff self-isolating after exposure to SARS-CoV-2 (35%); this led to spending less time with patients, possibly compromising care quality. Reported insufficient access to personal protective equipment (PPE) ranged from 12% for gloves to 32% for N-95 masks. Nonetheless, wearing PPE was tiresome, time-consuming, and presented potential communication barriers with patients. 58% of respondents reported higher stress levels, mainly related to lack of access to information or to rapidly changing guidelines. Respondents noted a negative financial impact—a decrease in income (70% among respondents from low-income countries) concurrently with increased personal expenditures (medical supplies, transportation, and PPE). Negative physical, psychological and economic impacts of COVID-19 on maternal and newborn healthcare providers were ongoing throughout 2020, especially in low-income countries. This can have

cannot be completely de-identified without removing key variables such as country, cadre, facility level, facility sector, area type and all the open-ended questions. This de-identification would limit the value of the dataset, making any replication of the analysis impossible. Data requests can be sent to the study PI Prof Lenka Benova at lbenova@itg.be and the ethics committee at the Institute of Tropical Medicine at irb@itg.be.

**Funding:** This study was funded by the Institute of Tropical Medicine's COVID-19 Pump Priming fund supported by the Flemish Government, Science & Innovation and by the Embassy of the United Kingdom in Belgium. LB is funded in part by the Research Foundation – Flanders (FWO) as part of her Senior Postdoctoral Fellowship. The funders had no role in study design, data collection and analysis, decision to publish, or preparation of the manuscript.

**Competing interests:** The authors have declared that no competing interests exist.

severe consequences for provision and quality of essential care. There is need to increase focus on the implementation of interventions aiming to support healthcare providers, particularly those in low- and middle-income countries to protect essential health services from disruption.

## Introduction

The novel Coronavirus disease (COVID-19) is a significant threat for human security and the attainment of health development goals. Its rapid spread led to the World Health Organization (WHO) pandemic declaration on March 11, 2020, three months after the first reported case. By August 2021, more than 215 million total confirmed cases and 4 million deaths had been reported worldwide [1]. The overwhelming burden of the disease and of measures to mitigate its spread (lockdowns with resulting transport disruptions, etc.) have overstretched healthcare systems' capacities in most countries, and caused detrimental effects on healthcare providers, including the risk of physical and mental/emotional disease [2–5].

The COVID-19 pandemic is devastating the world due to the lack of health systems' preparedness, and the neglect of its warning signs [6]. In 2003, a Severe Acute Respiratory Syndrome (SARS) epidemic emerged in China and rapidly spread to 29 Asian countries within six months, causing 8,096 cases and 774 deaths, with many healthcare providers being infected [7]. Subsequent epidemics within the last 18 years, included: H1N1 influenza (2009), Ebola in West Africa (2014–16), Zika (2016), and notably another coronavirus Middle-East Respiratory Syndrome (2017) [6, 8]. Healthcare providers' inequitable vulnerability within non-resilient health systems has been highlighted in the on-going COVID-19 pandemic. Occupational exposure to SARS-coronavirus-2 (SARS-CoV-2) infection has led to more than 152,888 confirmed cases and 1,412 deaths among healthcare providers worldwide [9]. Additionally, studies conducted during the first wave of the pandemic showed that healthcare providers were at increased risk of physical attacks, psychological harm and significant stressors [10–16]. Lai et al. reported high levels of depression (50%) and distress (72%) among healthcare providers in China where the disease first emerged [17]. The pandemic has exacerbated existing high workloads among healthcare providers with resulting fatigue and increased risk of burnout [11, 18, 19]. In France, among 1,025 medical doctors surveyed nationwide, 49% reported burnout during the pandemic [20]. Shortage of personal protective equipment (PPE), resulting in reduced feeling of protection in the workplace, has added pressure [12, 15, 21–24]. Some healthcare providers have rapidly adapted to requirements for home-based working, whilst others have faced lockdown restrictions affecting capacity to travel to work [25].

Gains in maternal and newborn health over the last few decades are at risk of being lost and maternal mortality and stillbirth rates rose during the COVID-19 pandemic [26]. Protecting health services for women and newborns is essential and cannot be rescheduled or postponed. In addition to continuing to provide routine maternal and newborn healthcare services, the workforce has also had to adapt to providing care to those infected with SARS-CoV-2. Maternal and neonatal healthcare providers–midwives, nurses, medical doctors (including medical officers and clinical officers) and the community health workforce–are the largest group within the global healthcare workforce and one of the professional groups most vulnerable to risks associated with the COVID-19 pandemic [27–30]. Safe-guarding their health and wellbeing is vital to protect essential services for women and children and enable health systems' response to the pandemic-related emerging challenges.

The lack of real-time data on the impact of COVID-19 on maternal and newborn health-care providers hinders health systems' capacities to evaluate its effectiveness and ability to protect and support this workforce. The effect of interventions undertaken in the first period of the pandemic to mitigate occupational risk—including development and implementation of training and safety guidelines, production and distribution of PPE during subsequent waves of the COVID-19 pandemic—remains unknown [31, 32]. The unknown economic impact of the pandemic on healthcare providers may increase stress and further affect the wellbeing and motivation of healthcare providers to respond to this rapidly changing pandemic.

Few studies during the first year of the pandemic explored the impact of COVID-19 on maternal and newborn healthcare providers [12, 14, 22, 28, 33] and only two included multi-national data [12, 14]. This study aims to describe maternal and newborn healthcare providers' experiences during the COVID-19 pandemic, and specifically to assess its physical, psychological and economic impacts from July to December 2020, after the initial wave of the pandemic.

## Methods

### Study design

This study used data collected during repeated rounds of a cross-sectional online global survey of maternal and newborn healthcare providers during the ongoing COVID-19 pandemic. Details about the study design and sampling have been published [12]. This paper presents findings from the second-round of the survey collected between July 5, 2020 and December 14, 2020. We focus on the challenges that maternal and newborn healthcare providers reported they faced in continuing to provide care to women and newborns after the initial wave of COVID-19, as well as measures to protect their wellbeing. Healthcare providers invited to participate in the survey included midwives, nurses, obstetricians/gynaecologists, neonatologists and paediatricians, among others. The survey was widely disseminated through international and national professional networks, social media channels (e.g., Twitter, Facebook, WhatsApp groups), and personal contacts. We additionally invited healthcare providers who responded to the first-round survey and agreed to participate in additional survey rounds.

### Questionnaire

The global survey research study team adapted the second-round survey questions based on first round responses and pertinent issues arising as the pandemic developed. This study team were multi-disciplinary and included health professionals, health systems experts, maternal and perinatal health experts, epidemiologists and public health researchers, acknowledged in a previously published commentary [34] and paper based on the first round of the global survey [12].

We maintained the core modules of the first-round questionnaire by asking respondents about their professional background, facility preparedness for and response to COVID-19 pandemic, and their own work-related experience in the month preceding the time they answered the survey. We collected data in the first section of the questionnaire to describe the sample characteristics (country, gender, job or type of professional categories, position held in health facility, type of healthcare services provided etc.). In this paper, we summarize findings from the questionnaire module on healthcare providers' experiences during the pandemic. Four dimensions were assessed using a mix of questions with multiple choice and open-text response options: 1) work-related experiences and physical impacts on healthcare providers (ability to reach their workplace, changes in staffing levels, exposure to aggressive behaviour as part of the job for any reason related to COVID-19, access to PPE, feeling safe in the workplace); 2) psychological impacts of COVID-19 on providers (perceived stress levels, feeling

valued by their community, access to formal mental health and psychological support, etc.); and 3) economic impact represented as change in income levels. We also use data from an open-text question in which respondents were asked to share their top three concerns regarding their ability to provide care to women and newborns during the COVID-19 pandemic.

The questionnaire was open online using Kobo Toolbox's online data collection feature [35], to allow use of built-in data collection quality checks in the questionnaire, including automated skip-patterns and answer restrictions when relevant. Round 2 was available in 11 languages (English, French, Arabic, Italian, Portuguese, Spanish, Japanese, German, Dutch, Russian, and Kiswahili). The questions analysed in this paper are listed in S1 Appendix, and the complete English-language questionnaire is available on the study website [36].

## Data processing and analysis

We analysed 1,405 submitted survey responses and cleaned the data by removing duplicate submissions (n = 16), lack of consent (n = 138) and submissions from respondents who were not currently practicing as clinical maternal and newborn healthcare providers such as nursing educators, public health officials, enumerators and administrators (n = 5). Additionally, submissions with more than 77% of questions with missing answers (n = 55) were removed from the dataset; this cut-off was chosen because it corresponds to the proportion of variables skipped when respondents drop-out after answering only the first section of the questionnaire (background characteristics module). Among the included respondents, 551 provided open-text responses to one or more of the variables of interest to this paper. The country income level was added to the database using the World Bank classification of the worlds' economies (according to 2020 gross national income) [37].

Quantitative and qualitative analyses were done in parallel for a convergent mixed-methods design. Quantitative analysis involved conducting descriptive statistics (frequencies and percentages) using Stata/SE version 16. Given the non-proportional sampling, we chose to disaggregate responses from three country income levels (high-income, middle-income, low-income countries), and report any variations in results descriptively. We applied thematic content analysis to analyse the responses to open-text questions. Each open-text response was independently coded by two researchers manually (DK and AS). The codes were discussed with the research team and grouped into themes. Qualitative data were further analysed to identify and document connections between the themes. The findings from the quantitative and qualitative strands were integrated at the data interpretation stage; quotes from the open-text responses were used to illustrate identified themes.

## Ethics

This study was approved by the Institutional Review Board at the Institute of Tropical Medicine in Antwerp Belgium, number 1372/20. Respondents provided informed consent online by checking a box affirming that they voluntarily agreed to participate in the survey. All data were collected anonymously.

## Results

### Characteristics of respondents

The final sample included 1,191 maternal and newborn healthcare providers from 77 countries (Table 1, and S1 Table). Most respondents were from middle-income countries (66%), followed by high-income (27%) and low-income countries (7%). Over half of pen-ended answers where from middle-income countries (51%), followed by high-income countries (35%) and

**Table 1. Characteristics of maternal and newborn care providers (n = 1,191).**

| Characteristic | High-income country (n = 317, 27%) | Middle-income country (n = 786, 66%) | Low-income country (n = 88, 7%) | Total (%) |
|---|---|---|---|---|
| **Gender** | | | | |
| Male | 49 (15.9) | 133 (17.1) | 60 (68.2) | 242 (20.6) |
| Female | 256 (82.8) | 641 (82.2) | 28 (31.8) | 925 (78.6) |
| Prefer not to say | 4 (1.3) | 6 (0.8) | 0 (0) | 10 (0.8) |
| **Job** | | | | |
| Midwife | 130 (41.4) | 91 (11.7) | 6 (6.9) | 227 (19.3) |
| Nurse-midwife | 29 (9.2) | 30 (3.9) | 7 (8) | 66 (5.6) |
| Nurse | 55 (17.5) | 296 (38.2) | 9 (10.3) | 360 (30.6) |
| Obstetrician/Gynaecologist | 60 (19.1) | 173 (22.3) | 16 (18.4) | 249 (21.2) |
| Neonatologist | 15 (4.8) | 21 (2.7) | 0 (0) | 36 (3.1) |
| Paediatrician | 15 (4.8) | 29 (3.7) | 1 (1.1) | 45 (3.8) |
| Medical doctor | 6 (1.9) | 110 (14.2) | 39 (44.8) | 155 (13.2) |
| Other | 4 (1.3) | 25 (3.2) | 9 (10.3) | 38 (3.2) |
| **Position** | | | | |
| Head of facility | 15 (4.8) | 20 (2.7) | 14 (16.9) | 49 (4.3) |
| Head of department or ward | 32 (10.3) | 86 (11.5) | 21 (25.3) | 139 (12.2) |
| Head of team | 41 (13.2) | 47 (6.3) | 12 (14.5) | 100 (8.8) |
| Team member | 164 (52.7) | 360 (48.2) | 26 (31.3) | 550 (48.2) |
| Locum or interim member | 5 (1.6) | 74 (9.9) | 0 (0) | 79 (6.9) |
| Independent or self-practicing | 50 (16.1) | 68 (9.1) | 6 (7.2) | 124 (10.9) |
| Other | 4 (1.3) | 92 (12.3) | 4 (4.8) | 100 (8.8) |
| **Facility type** | | | | |
| Public (national) | 97 (31.4) | 467 (60.5) | 31 (35.6) | 595 (50.9) |
| Public (university or teaching) | 43 (13.9) | 60 (7.8) | 13 (14.9) | 116 (9.9) |
| Public (district level or below) | 61 (19.7) | 113 (14.6) | 7 (8) | 181 (15.5) |
| Social security | 1 (0.3) | 1 (0.1) | 0 (0) | 2 (0.2) |
| Health insurance | 4 (1.3) | 36 (4.7) | 0 (0) | 40 (3.4) |
| Private | 36 (11.7) | 19 (2.5) | 13 (14.9) | 68 (5.8) |
| Non-governmental | 1 (0.3) | 10 (1.3) | 7 (8) | 18 (1.5) |
| Faith-based or mission | 2 (0.6) | 8 (1) | 6 (6.9) | 16 (1.4) |
| Independent/self-practicing | 51 (16.5) | 16 (2.1) | 3 (3.4) | 70 (6) |
| Other | 13 (4.2) | 42 (5.4) | 7 (8) | 62 (5.3) |
| **Area type** | | | | |
| Large city | 81 (26.7) | 283 (37.2) | 44 (51.8) | 408 (35.5) |
| Small city | 103 (34) | 149 (19.6) | 22 (25.9) | 274 (23.9) |
| Town | 77 (25.4) | 78 (10.3) | 2 (2.4) | 157 (13.7) |

*(Continued)*

**Table 1.** (Continued)

| Characteristic | High-income country (n = 317, 27%) | Middle-income country (n = 786, 66%) | Low-income country (n = 88, 7%) | Total (%) |
|---|---|---|---|---|
| Village/Rural area | 37 (12.2) | 207 (27.2) | 14 (16.5) | 258 (22.5) |
| Other | 5 (1.7) | 43 (5.6) | 3 (3.5) | 51 (4.5) |
| **Healthcare service provided** | | | | |
| *(more than one option possible)* | | | | |
| Antenatal care | 186 (60.4) | 343 (51.6) | 48 (56.5) | 577 (54.5) |
| Intrapartum care | 193 (62.7) | 217 (32.6) | 52 (61.2) | 462 (43.7) |
| Postnatal care | 207 (67.2) | 281 (42.3) | 51 (60) | 539 (50.9) |
| Outpatient breastfeeding support | 94 (30.5) | 160 (24.1) | 22 (25.9) | 276 (26.1) |
| Neonatal care for small and sick newborns | 39 (12.7) | 113 (17) | 25 (29.4) | 177 (16.7) |
| Surgical care | 54 (17.5) | 107 (16.1) | 28 (32.9) | 189 (17.9) |
| Family planning provision or counselling | 70 (22.7) | 164 (24.7) | 35 (41.2) | 269 (25.4) |
| Abortion and post-abortion care | 82 (26.6) | 117 (17.6) | 30 (35.3) | 229 (21.6) |
| Home care or community outreach | 85 (27.6) | 181 (27.2) | 22 (25.9) | 288 (27.2) |
| **Provides care in** | | | | |
| One facility | 215 (69.8) | 575 (74.4) | 39 (44.8) | 829 (71) |
| Multiple facilities | 93 (30.2) | 198 (25.6) | 48 (55.2) | 339 (29) |
| **Healthcare facility level** | | | | |
| Referral hospital | 84 (27.2) | 107 (13.9) | 34 (39.5) | 225 (19.3) |
| District/regional hospital | 95 (30.7) | 130 (16.9) | 6 (7) | 231 (19.9) |
| Health centre | 42 (13.6) | 48 (6.3) | 8 (9.3) | 98 (8.4) |
| Polyclinic or clinic | 14 (4.5) | 279 (36.3) | 14 (16.3) | 307 (26.4) |
| Health post/unit or dispensary | 1 (0.3) | 11 (1.5) | 0 (0) | 12 (1.1) |
| Birth centre | 14 (4.5) | 110 (14.3) | 1 (1.2) | 125 (10.7) |
| Home-based care | 2 (0.6) | 3 (0.4) | 0 (0) | 5 (0.4) |
| Independent/self-practicing | 45 (14.6) | 25 (3.3) | 7 (8.1) | 77 (6.6) |
| Other | 12 (3.9) | 55 (7.2) | 16 (18.6) | 83 (7.1) |

[*] HIC = High-income countries, MIC = Middle-income countries, LIC = Low-income countries

low-income countries (14%). Overall, most respondents were female (79%), whereas in low-income countries, most respondents were male (68%). Nurses (31%), midwives/nurse-midwives (25%), and obstetricians/gynaecologists (21%) were the three most common cadres in our sample. The largest group of respondents from high-income countries were midwives (41%), from middle-income countries nurses (38%), and from low-income countries were medical doctors (45%). The majority of respondents in all three country income groups worked in the public sector (76%), and in cities (59%).

Services provided by respondents were antenatal (54%), intrapartum (44%) and postnatal care (51%). Overall, 29% of respondents worked in multiple health facilities, and this was higher (55%) among respondents from low-income countries. Respondents worked in poly-clinics (26%) followed by district/regional (20%), and referral (19%) hospitals. Almost one third of respondents from high-income countries worked in district/regional hospitals (31%) while 36% of respondents in middle-income countries worked in polyclinics or clinics and 39% in referral hospitals among respondents from low-income countries.

Findings are summarised as three main themes including 1) work-related experiences and physical impact, 2) psychological impact, and 3) economic impact of the COVID-19 pandemic on maternal and newborn healthcare providers.

## Work-related experiences and physical impact for maternal and newborn healthcare provider's during the COVID-19 pandemic

Findings on experiences regarding resources, infrastructure and staffing during the COVID-19 pandemic are shown in Table 2. Nearly one third of respondents reported a decrease in staffing levels during the month preceding the survey. Leading reasons included changes in staff rotation systems (53%) and self-isolation following exposure to SARS-CoV-2 (35%). Healthcare providers' health-related issues such as COVID-19 illness (26%) and burnout

**Table 2. Maternal and newborn healthcare providers' experiences with resources, infrastructure and staffing during the COVID-19 pandemic in the month preceding the survey (n = 1,191).**

|  | HIC (n = 317, 27%) | MIC (n = 786, 66%) | LIC (n = 88, 7%) | Total (%) |
|---|---|---|---|---|
| **Staffing level** |  |  |  |  |
| Staffing not affected | 199 (63.8) | 218 (30.1) | 38 (45.8) | 455 (40.6) |
| Staffing levels decreased | 87 (27.9) | 199 (27.4) | 33 (39.8) | 319 (28.5) |
| Staffing levels increased | 18 (5.8) | 176 (24.3) | 9 (10.8) | 203 (18.1) |
| Don't know | 8 (2.6) | 132 (18.2) | 3 (3.6) | 143 (12.8) |
| **Reasons for decrease in staffing level (multiple answers allowed; n = 315; answered by those who reported a decrease in staffing levels)** |  |  |  |  |
| Change in staff rotation or shift schedule | 28 (32.2) | 115 (59) | 24 (72.7) | 167 (53) |
| Staff unable to reach workplace | 4 (4.6) | 34 (17.4) | 18 (54.5) | 56 (17.8) |
| Staff re-assigned to COVID-19 wards | 22 (25.3) | 66 (33.8) | 4 (12.1) | 92 (29.2) |
| Staff isolating following exposure to COVID-19 | 29 (33.3) | 75 (38.5) | 7 (21.2) | 111 (35.2) |
| Staff ill with COVID-19 | 23 (26.4) | 56 (28.7) | 5 (15.2) | 84 (26.7) |
| Staff off due to childcare | 30 (34.5) | 18 (9.2) | 2 (6.1) | 50 (15.9) |
| Staff off due to stress or burnout | 26 (29.9) | 22 (11.3) | 10 (30.3) | 58 (18.4) |
| Don't know | 4 (4.6) | 12 (6.2) | 0 (0) | 16 (5.1) |
| **Ability to reach workplace (n = 1,116)** |  |  |  |  |
| I can reach my workplace easily | 295 (94.2) | 465 (64.5) | 55 (67.1) | 815 (73) |
| I can reach my workplace with some difficulty | 13 (4.2) | 154 (21.4) | 23 (28) | 190 (17) |
| It is very difficult for me to reach the workplace | 3 (1) | 68 (9.4) | 4 (4.9) | 75 (6.7) |
| It is impossible for me to reach the workplace | 2 (0.6) | 34 (4.7) | 0 (0) | 36 (3.2) |
| **Reasons for difficulty in reaching workplace (multiple answers allowed; n = 295)** |  |  |  |  |

*(Continued)*

**Table 2.** (*Continued*)

| | HIC (n = 317, 27%) | MIC (n = 786, 66%) | LIC (n = 88, 7%) | Total (%) |
|---|---|---|---|---|
| Lockdown measures | 3 (18.8) | 58 (23.0) | 6 (22.2) | 67 (22.7) |
| Curfew | 2 (12.5) | 34 (13.5) | 7 (25.9) | 43 (14.6) |
| Public transportation availability | 5 (31.3) | 157 (62.3) | 19 (70.4) | 181 (61.4) |
| Other | 8 (50.0) | 41 (16.3) | 4 (14.8) | 53 (18.0) |
| **Sufficient gloves** | | | | |
| No | 15 (4.8) | 93 (13) | 26 (30.6) | 134 (12.1) |
| Yes | 292 (93.9) | 605 (84.9) | 56 (65.9) | 953 (85.9) |
| Not required | 4 (1.3) | 15 (2.1) | 3 (3.5) | 22 (2) |
| **Sufficient N-95 masks** | | | | |
| No | 78 (25.7) | 214 (31.1) | 56 (69.1) | 348 (32.4) |
| Yes | 198 (65.3) | 450 (65.3) | 20 (24.7) | 668 (62.3) |
| Not required | 27 (8.9) | 25 (3.6) | 5 (6.2) | 57 (5.3) |
| **Sufficient surgical masks** | | | | |
| No | 32 (10.6) | 164 (24) | 30 (35.3) | 226 (21.1) |
| Yes | 256 (85) | 496 (72.6) | 50 (58.8) | 802 (75) |
| Not required | 13 (4.3) | 23 (3.4) | 5 (5.9) | 41 (3.8) |
| **Sufficient face/eye protection** | | | | |
| No | 66 (21.5) | 193 (27.9) | 64 (75.3) | 323 (29.8) |
| Yes | 223 (72.6) | 474 (68.5) | 15 (17.6) | 712 (65.7) |
| Not required | 18 (5.9) | 25 (3.6) | 6 (7.1) | 49 (4.5) |
| **Sufficient aprons** | | | | |
| No | 45 (15.4) | 165 (26.3) | 40 (49.4) | 250 (25) |
| Yes | 228 (77.8) | 424 (67.5) | 36 (44.4) | 688 (68.7) |
| Not required | 20 (6.8) | 39 (6.2) | 5 (6.2) | 64 (6.4) |
| **Sufficient PPE to change between patients** | | | | |
| No | 91 (29.3) | 252 (34.2) | 75 (85.2) | 418 (36.8) |
| Yes | 197 (63.3) | 441 (59.9) | 10 (11.4) | 648 (57.1) |
| Not required | 23 (7.4) | 43 (5.8) | 3 (3.4) | 69 (6.1) |
| **Faced challenges with personal protective equipment** | 74 (24) | 215 (29.7) | 52 (62.7) | 341 (30.6) |
| **Possible to get tested for COVID-19 as a healthcare provider** | | | | |
| No | 21 (6.7) | 50 (6.7) | 25 (28.4) | 96 (8.3) |
| Yes, regardless of symptoms or exposure | 82 (26.1) | 278 (37) | 19 (21.6) | 379 (32.9) |
| Yes, only if exposed to COVID-19 suspected/confirmed cases | 107 (34.1) | 223 (29.7) | 26 (29.5) | 356 (30.9) |
| Yes, only if symptomatic | 87 (27.7) | 143 (19) | 16 (18.2) | 246 (21.3) |
| Don't know | 17 (5.4) | 57 (7.6) | 2 (2.3) | 76 (6.6) |

(*Continued*)

**Table 2.** (Continued)

| | HIC (n = 317, 27%) | MIC (n = 786, 66%) | LIC (n = 88, 7%) | Total (%) |
|---|---|---|---|---|
| **Cost of test (n = 974; answered only by those who reported that they can get tested for COVID-19)** | | | | |
| Free of charge | 218 (79.3) | 514 (80.6) | 55 (90.2) | 787 (80.8) |
| Paid by healthcare provider | 17 (6.2) | 62 (9.7) | 5 (8.2) | 84 (8.6) |
| Other | 4 (1.5) | 4 (0.6) | 0 (0) | 8 (0.8) |
| Don't know | 36 (13.1) | 58 (9.1) | 1 (1.6) | 95 (9.8) |

* HIC = High-income countries, MIC = Middle-income countries, LIC = Low-income countries

(18%) were also highly reported to contribute to decreased staffing levels. Low-income country healthcare providers reported inability to reach the workplace was the main reason for decreased staffing (55%), whereas in high- and middle-income countries, about one third of respondents reported providers' re-assignment from maternal newborn wards to provide care in COVID-19 wards. Difficulty in reaching the workplace during the month before the survey was experienced by nearly 25% of respondents, and this proportion was lower among respondents from high-income countries (6%) compared to low- and middle-income countries (33% and 35%, respectively). Leading causes were unavailability of public transportation (61%), lockdown measures (22%), and curfews (15%).

Workplace PPE challenges were common, experienced by 24% of respondents from high-income and 64% from low-income countries. Rates of access differed according to the PPE component, ranging from 12% reporting insufficient gloves, to 32% reporting insufficient N-95 masks. One third of respondents reported that they did not have sufficient PPE to change between patients. The proportion of respondents reporting insufficient PPE was much higher in low-income countries compared to high- and middle-income countries across all types of equipment asked about in the questionnaire. Overall, SARS-CoV-2 polymerase chain reaction (PCR) tests were mostly available and free of charge in the month preceding the survey, however, almost one third of respondents from low-income countries did not have any access to PCR testing, compared to 7% of respondents from high- and middle-income countries (Table 2).

Findings on other experiences in the workplace are summarised in Table 3. More than one quarter of respondents (28%) reported experiencing aggressive behaviours in their workplace during the month preceding the survey, mostly verbal aggression/shouting (57%). These aggressive behaviours reportedly targeted healthcare providers themselves (55%) and/or their colleagues (44%), having been most commonly perpetrated by patients (36%) and/or patients' families (40%). In low-income countries, colleagues and public or government officials were perpetrators of aggressive behaviours in 22% of the cases each. One third of respondents reported not having access to formal mental health support during the month preceding the survey, with large differences between low-income (61%) and high- and middle-income countries (30% and 33%, respectively). 44% of respondents stated that their concerns in general were well or completely addressed by their facility or professional organization.

## Psychological impact

Different types of the psychological impact of the COVID-19 pandemic on maternal and newborn healthcare providers during the month preceding the pandemic are shown in Table 4. Half of all respondents (55%) reported feeling well and completely protected from SARS-CoV-

**Table 3. Maternal and newborn healthcare providers' experiences with aggression and support from the workplace during the COVID-19 pandemic in the month preceding the survey (n = 1,191).**

| | HIC (n = 317, 27%) | MIC (n = 786, 66%) | LIC (n = 88, 7%) | Total (%) |
|---|---|---|---|---|
| **Exposure to aggressive behaviour in the workplace** | 59 (18.8) | 227 (31.9) | 18 (22.5) | 304 (27.5) |
| **Type of aggressive behaviour (multiple answers allowed; n = 288)** | | | | |
| Animosity or discrimination | 28 (49.1) | 73 (34.3) | 1 (5.6) | 102 (35.4) |
| Harassment | 9 (15.8) | 22 (10.3) | 2 (11.1) | 33 (11.5) |
| Verbal aggression, shouting | 38 (66.7) | 117 (54.9) | 9 (50) | 164 (56.9) |
| Intimidation / threats | 15 (26.3) | 56 (26.3) | 6 (33.3) | 77 (26.7) |
| Threatening gestures, including with a weapon or a dangerous object | 2 (3.5) | 11 (5.2) | 0 (0) | 13 (4.5) |
| Physical violence (including shoving, punching, kicking, etc.) | 3 (5.3) | 12 (5.6) | 2 (11.1) | 17 (5.9) |
| Spitting or coughing | 2 (3.5) | 26 (12.2) | 3 (16.7) | 31 (10.8) |
| Sexual violence | 0 (0) | 3 (1.4) | 0 (0) | 3 (1) |
| Self-harm | 2 (3.5) | 4 (1.9) | 0 (0) | 6 (2.1) |
| **Target of the aggressive behaviour (multiple answers allowed; n = 289)** | | | | |
| Healthcare providers themselves | 42 (73.7) | 112 (52.3) | 6 (33.3) | 160 (55.4) |
| Colleagues | 31 (54.4) | 88 (41.1) | 7 (38.9) | 126 (43.6) |
| Family members | 1 (1.8) | 8 (3.7) | 4 (22.2) | 13 (4.5) |
| Friends or relatives | 1 (1.8) | 7 (3.3) | 6 (33.3) | 14 (4.8) |
| Patients | 11 (19.3) | 51 (23.8) | 1 (5.6) | 63 (21.8) |
| Aggression toward objects (desk, wall, etc) | 2 (3.5) | 14 (6.5) | 3 (16.7) | 19 (6.6) |
| **Perpetrator of the aggressive behaviour (multiple answers allowed; n = 288)** | | | | |
| Healthcare providers themselves | 1 (1.8) | 33 (15.5) | 1 (5.6) | 35 (12.2) |
| Colleagues | 8 (14) | 23 (10.8) | 4 (22.2) | 35 (12.2) |
| Family members | 0 (0) | 8 (3.8) | 3 (16.7) | 11 (3.8) |
| Friends or relatives | 1 (1.8) | 7 (3.3) | 3 (16.7) | 11 (3.8) |
| Community member (e.g., neighbour or teacher) | 5 (8.8) | 24 (11.3) | 3 (16.7) | 32 (11.1) |
| Patient | 28 (49.1) | 74 (34.7) | 2 (11.1) | 104 (36.1) |
| Patient's family | 34 (59.6) | 76 (35.7) | 5 (27.8) | 115 (39.9) |
| Stranger | 7 (12.3) | 26 (12.2) | 2 (11.1) | 35 (12.2) |
| Public or government official | 3 (5.3) | 22 (10.3) | 4 (22.2) | 29 (10.1) |
| **Access to formal mental health support (n = 1,104)** | | | | |
| No access | 93 (29.5) | 232 (32.8) | 50 (61) | 375 (34) |
| Available, but not free of charge | 57 (18.1) | 50 (7.1) | 10 (12.2) | 117 (10.6) |
| Free access covered by facility/organisation | 112 (35.6) | 251 (35.5) | 18 (22) | 381 (34.5) |
| Don't know | 53 (16.8) | 174 (24.6) | 4 (4.9) | 231 (20.9) |
| **Concerns addressed by facility or by professional organisation (Likert scale, n = 1,085)** | | | | |
| Not at all | 23 (7.4) | 63 (9.1) | 19 (23.8) | 105 (9.7) |
| Minimally | 50 (16.1) | 88 (12.7) | 24 (30) | 162 (14.9) |
| Somewhat | 111 (35.8) | 195 (28.1) | 29 (36.3) | 335 (30.9) |
| Well | 100 (32.3) | 224 (32.2) | 8 (10) | 332 (30.6) |
| Completely | 26 (8.4) | 125 (18) | 0 (0) | 151 (13.9) |

* HIC = High-income countries, MIC = Middle-income countries, LIC = Low-income countries

2 infection risk in their workplace, but only 14% of respondents from low-income countries reported this. About 30% of respondents felt that they were valued very little or not at all by their community. More than half of respondents (58%) reported higher stress levels during the survey period compared with the start of the pandemic, consistently across the country income groups.

**Table 4. The psychological impact of the COVID-19 pandemic on maternal and newborn healthcare providers during the month preceding the survey (n = 1,191).**

|  | HIC (n = 317, 27%) | MIC (n = 786, 66%) | LIC (n = 88, 7%) | Total (%) |
|---|---|---|---|---|
| **Feeling protected in the workplace (Likert scale)** |  |  |  |  |
| Not at all | 6 (2) | 40 (5.5) | 12 (13.6) | 58 (5.2) |
| Minimal protection | 19 (6.3) | 81 (11.1) | 28 (31.8) | 128 (11.4) |
| Some protection | 103 (34) | 188 (25.8) | 35 (39.8) | 326 (29.1) |
| Well protected | 137 (45.2) | 250 (34.3) | 12 (13.6) | 399 (35.6) |
| Completely protected | 38 (12.5) | 170 (23.3) | 1 (1.1) | 209 (18.7) |
| **Feeling valued by community (Likert scale)** |  |  |  |  |
| Not at all | 34 (11) | 53 (7.6) | 15 (18.3) | 102 (9.4) |
| Very little | 59 (19.1) | 138 (19.9) | 18 (22) | 215 (19.8) |
| Somewhat | 133 (43) | 267 (38.5) | 27 (32.9) | 427 (39.4) |
| Highly | 77 (24.9) | 177 (25.5) | 15 (18.3) | 269 (24.8) |
| Unsure/don't know | 6 (1.9) | 58 (8.4) | 7 (8.5) | 71 (6.5) |
| **Stress levels compared to beginning of outbreak** |  |  |  |  |
| Substantially lower | 20 (6.5) | 49 (7.1) | 7 (8.5) | 76 (7) |
| Somewhat lower | 58 (18.7) | 100 (14.5) | 16 (19.5) | 174 (16.1) |
| Same as the beginning of the outbreak | 65 (21) | 127 (18.4) | 9 (11) | 201 (18.6) |
| Somewhat higher | 129 (41.6) | 259 (37.5) | 38 (46.3) | 426 (39.4) |
| Substantially higher | 38 (12.3) | 155 (22.5) | 12 (14.6) | 205 (18.9) |

\* HIC = High-income countries, MIC = Middle-income countries, LIC = Low-income countries

## Economic impact of the COVID-19 pandemic

Almost 30% of respondents reported a decrease in their income during the month preceding the survey compared to before the COVID-19 pandemic. This proportion reached 70% among respondents from low-income countries (Fig 1). Open-text responses indicated providers

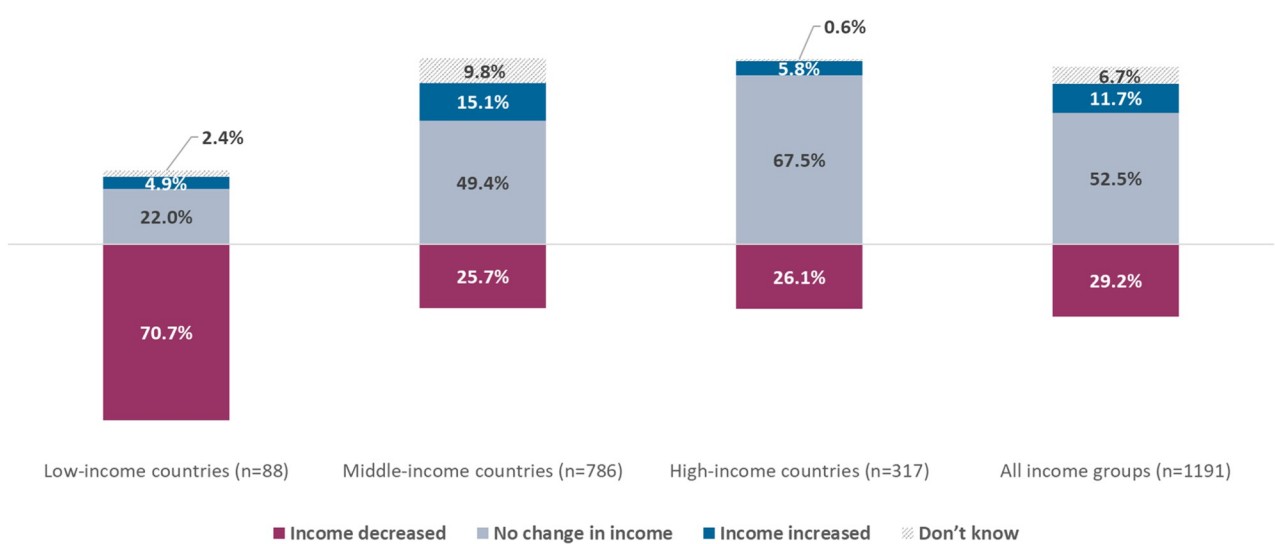

**Fig 1. Percentage of maternal and newborn healthcare providers reporting changes in their income during the month preceding the survey compared to before the COVID-19 pandemic, by country income group (n = 1,191).**

working in the private sector were more affected, as this decrease in income was linked to lower clinic patient attendance due to increases in costs associated with care-seeking (public transport, additional charges for PPE etc.), and fear of getting infected in healthcare facilities. Some respondents also explained that they have incurred additional costs related to providing care as medical supplies', PPEs' and transportations' "*prices have skyrocketed*".

## Connections between dimensions of the impact of COVID-19 and provision of maternal and newborn quality care

Findings from the quantitative and qualitative data are summarised in Fig 2 according to three levels: a) Experiences related to health systems, facilities and teams (classified as pre-existing issues, COVID-19 specific issues, or both); b) Impact on healthcare providers (physical, psychological and economic); and c) Impact on women, newborns and families. Five key connections (Fig 2 coloured lines) between the factors affecting healthcare providers during the COVID-19 pandemic were identified across the three levels, including how these impacted healthcare provision to women and newborns.

The first connection (yellow colour connection line) represents the consequences of structural and health system-related issues on the physical, psychological and economic impacts on healthcare providers, and on the care provided to women and newborns. Frequent stock-outs of protective equipment, including facemasks, contributed to healthcare providers feeling unsafe in the workplace (psychological effect) and being unsafe (physically unsafe). Pre-existing issues such as the lack of space and adequate ventilation in health facilities further challenged infection prevention and control measures, and the lack of COVID-19 testing for women attending health facilities for care also contributed to fear and feeling of unsafety among healthcare providers. A midwife in Lebanon wrote: "*[. . .] women who come to give birth are not tested, [I am concerned] whether they are positive asymptomatic, and the precautions are the same as before without the use of more equipment for midwives to be more protected*". This impacted healthcare provision, especially trust between patient and providers. For instance, some patients were reportedly "*withholding symptoms for fear of being labelled*

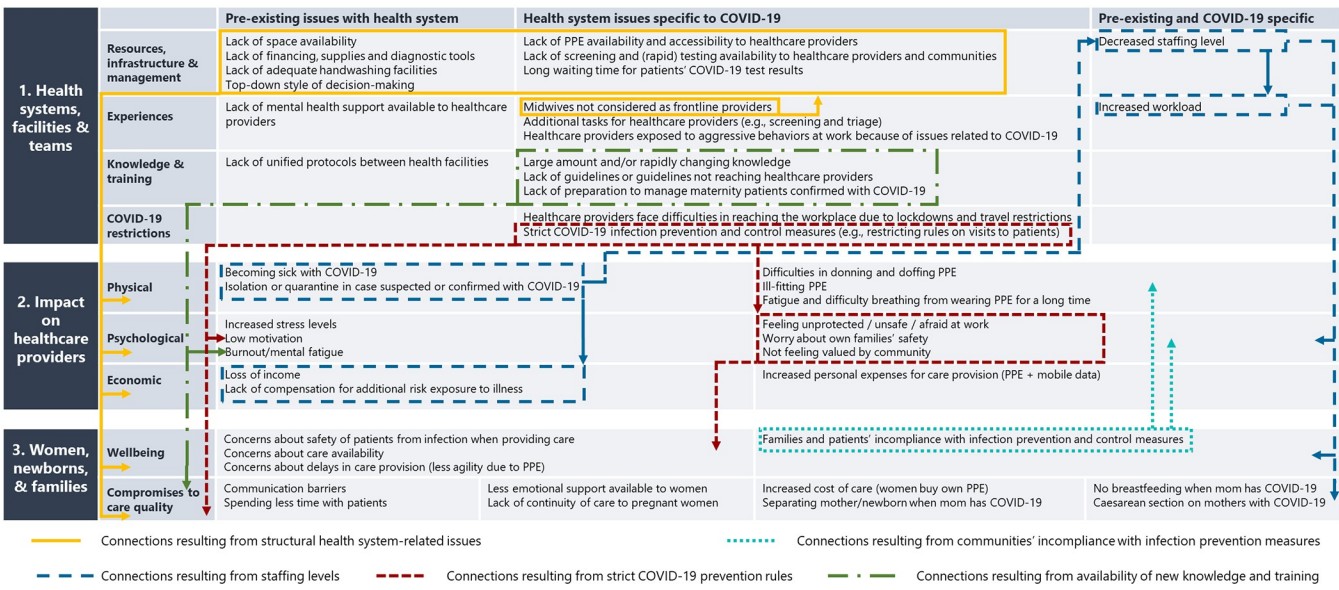

**Fig 2. Connections between dimensions of the impact of COVID-19 and provision of maternal and newborn quality care.**

*[turned away from care]"*, some providers feared to manage patients presenting with COVID-19 signs and symptoms: *"Women with fever due to other causes are being mismanaged or mistreated fearing [because of the fear to contract] COVID-19. This delays their care and may lead to disastrous consequences in some cases" (obstetrician/gynaecologist, Uganda)*.

The second connection (teal colour connection line) represents how patients' (and their families' or care givers') poor compliance with infection prevention and control measures also contributed to healthcare providers' concerns about safety and risk of infection in the workplace. A nurse from Zambia stated: *"General public still don't seem to have accepted that COVID-19 is there and real; they are not keen to taking preventive measures like wearing face masks and hand washing. This is a danger to the nurses and midwives."*

Third, staff shortages (blue colour connection line) were related to both physical, psychological and economic impacts of the COVID-19 pandemic. Some respondents mentioned that the necessity of self-isolation after COVID-19 infection led to *"decreased salary during necessary self-quarantine"*, as mentioned by an obstetrician/gynaecologist in Slovakia. Conversely, staff illness, quarantine and/or self-isolation meant that fewer healthcare providers were available to provide care, which led to increased workload for those who continued to provide care, thus leading to fatigue, burnout and raising concerns among healthcare providers about the availability of care for women and newborns, as well as compromising care quality from shortage of time with patients. *"[A]t the moment many doctors, midwives, nurses are sick with COVID-19 and the rest of the medical personnel who are not sick have to work and be on call instead. Many doctors, midwives are tired from the constant additional workload and of course this has a negative impact on the provision of care by tired health workers, it leads to some reduction in the quality of perinatal care"* (Obstetrician/ Gynaecologist from Uzbekistan).

Fourth, the strict COVID-19 precautions themselves affected healthcare providers physically and psychologically (red colour connection line). The constant need to wear PPE was challenging for healthcare providers for several reasons. A physician in the US noted feeling *"tiredness and personal fatigue when using PPE on a daily basis"*, while a female physician from the Democratic Republic of Congo complained about *"the predominance of ill-fitting/too big PPE in supplies"*. The time taken to properly "don and doff" PPE further compromised timely provision of care to women and newborns. Facemasks were in some cases considered as communication barriers, impeding the bonding between the providers and women. Many healthcare providers also reported women's dissatisfaction with the rules banning/reducing visitors in health facilities and perceived that this led to a reduction in the support available to women before, during and after childbirth (Box 1).

Fifth, availability of knowledge and training on COVID-19 in general, and on COVID-19 and maternity care, were linked to a psychological impact on healthcare providers, and to the quality of care provided to women and newborns (green colour connection line). Some healthcare providers reported that they suffered from mental "information" fatigue as a result of the large amount of information flow and rapidly changing guidelines. On the other hand, the lack of guidelines, or guidelines not reaching the healthcare providers were also causes of stress among healthcare providers, and influenced the quality of care. A midwife from Spain wrote: *"[there is] anger due to changing protocols, little information, no one is responsible for our preparation or emotional state"*. This reportedly contributed to the provision of non-evidence-based practices and impacted on the quality of care, particularly for women with confirmed COVID-19 such as separation of mother/newborn, banning breastfeeding, and birth by caesarean section. A few examples are illustrated in Box 1.

Box 1. Examples of the impact of the COVID-19 pandemic on the quality of care provided to women and newborns, reported as concerns of healthcare providers

### Poor emotional support to women

Some respondents reported a decrease in emotional support for women because of, among others, cancellation of individual and group counselling during antenatal and postnatal care, lack of adequate measures in addressing COVID-19 related fears of women, reduction of allowing families' support including companionship during labour and childbirth, and separation of mothers and their babies in maternity wards. Some respondents highlighted that the prohibition of spouses and relatives' attendance in delivery rooms has resulted in *"reduced support to women in the post-surgery period from a family member" (midwife, Cameroon)* or in some cases, in *"post-partum depression among women who felt abandoned during this important time in their lives" (Medical Doctor, France)*.

### Respect of women dignity

Respondents raised issues related to preservation of women's dignity and privacy during labour and childbirth. For instance, some reported the sharing of waiting rooms among women in labour and other patients. Other respondents also reported that women had little power to decide whether they wanted to breastfeed their babies or not or whether they wanted to stay with them or not. According to respondents, decisions were solely triggered by COVID-19 prevention standards of the health facility.

> *"The separation of babies from their [COVID-19 positives] mothers is recommended by the Spanish neonatal society and maternal breastfeeding is considered] an unsafe method. . . All this has led to the loss of women's right to decide." (midwife, Spain)*

### Dysfunctional referral system

Respondents perceived that the COVID-19 pandemic negatively impacted the functioning of referral systems. This, according to them, was caused by a lack of or poor communication among healthcare providers in the referral system.

> *"There is no feedback and follow up of the mothers who have been referred." (obstetrician/gynaecologist, India)*

## Discussion

In this study of maternal and newborn healthcare providers with responses from 77 countries, we found that nearly a third of providers experienced working conditions with insufficient levels of staffing, shortages of PPE, and decreased income and increased out-of-pocket expenditure related to work during the study period. These declines in health resources and income

were reported more commonly by providers from low-income countries compared to high- and middle-income countries. Staff shortages were mostly attributed to illness from COVID-19 or exposure to SARS-CoV-2 which required self-isolation, and re-assignment of some maternal and newborn healthcare providers to COVID-19 management units, more so by respondents from middle- and high-income countries compared to low-income countries. On the other hand, changes in staffing rotation patterns, and difficulties in reaching workplace due to restrictive measures such as lockdowns and curfews, were dominant in low-income countries. Moreover, almost half of providers reported increased stress levels and felt unsafe in their workplace during the study period as compared to before the COVID-19 pandemic. Our analysis revealed multiple interactions between providers' work-related experience during the pandemic, its physical, psychological, and economic impacts on them, and their ability to provide quality maternal and newborn healthcare. These interactions appeared to be exacerbated by pre-existing health system challenges, such as inadequate health infrastructure and resources.

Throughout 2020, countries implemented various restriction measures; evidence suggests that the highest stringency index was recorded in middle-income countries, followed by low- and high- income countries [38]. The period covered by our study (July to December 2020) represents a time beyond the initial emergency response to the pandemic. At this time, we would have expected some of the primary concerns regarding PPE, accessibility to guidelines on provision of care, and availability of diagnostic tools to have been largely resolved or addressed as part of planning for a long-term response. Nonetheless, our results show that challenges experienced at the onset of the pandemic persisted for many healthcare providers [12], particularly for those working in low-income countries, for many months. Many of these challenges are related to resource availability such as shortages of PPE, lack of testing availability for healthcare providers, and unavailability of formal mental health support. These results raise the issue of equitable access to COVID-19 health supplies (including PPE, tests, treatments, and vaccines); a central element to ending the acute phase of the COVID-19 pandemic and maintaining provision of essential healthcare services [32]. Low-income countries were faced with a weaker health-system infrastructure before the COVID-19 pandemic, including pre-existing shortage of skilled maternal and newborn healthcare providers [39, 40]. The potential collateral damage of the COVID-19 restrictive measures (e.g. lockdowns, curfews, unavailability of public transport) may exacerbate the deficit in health resources in these settings, and worsen maternal and newborn health outcomes. We therefore recommend prioritizing maternal and newborn healthcare providers, particularly in low-income countries, during the COVID-19 pandemic response, and future health system preparedness plans. This prioritization entails fulfilling providers' needs to continue working safely by providing them with adequate PPE, testing and screening infrastructure, and mental health support, as well as not reassigning them to crisis response duties.

Shortages of PPE for maternal and newborn providers were consistently reported in previous studies in Nigeria, Australia and Belgium, and a global survey of care providers to small and sick newborns [4, 41–43]. As maternal and newborn providers were not directly involved in care provision to COVID-19 patients in many settings, it is possible that PPE stocks were not being prioritised for them. However, compared with a global survey conducted during the early period of the COVID-19 pandemic [12], our study suggests an ongoing deficit of PPE for providers working in low-income countries. Many health facilities in low-income countries might have not replenished their PPE stockpile between the first and subsequent waves of the COVID-19 pandemic, and reasons for this may include the breakdown of global PPE supply chains, and resulting competition between countries and states [44].

Our findings revealed increased levels of stress among health providers, who are exposed to emotional fatigue and burnout resulting from long-working hours and high work burden in understaffed and unsafe environments, and a persistent fear of contracting the disease. These high levels of stress were noted in all three country income groups. In addition, many health providers reported being exposed to aggressive behaviours from patients and/or their families aggravating the stress during this period. These findings concur with those of several other works during the COVID-19 pandemic [17, 45–51]. Studies in Turkey and the United Kingdom documented that stress levels and anxiety increased among maternal and newborn healthcare providers during the pandemic [4, 52, 53]. In Nigeria, research showed high levels of stress resulting from increased workload, and the fear of stigmatisation after potential exposure to SARS-CoV-2 [43]. We note that a high proportion of respondents from middle- and high-income countries continued to report higher levels of stress, despite relatively low reporting of other challenges (shortage of PPE, shortage of staffing, absence of mental health support). This suggests that other factors may be underlying healthcare providers' experiences of stress during a pandemic, which were not analysed in our study, such as the level of restriction measures. Future research should explore pathways leading to the psychological wellbeing of maternal and newborn healthcare providers in the context of a health system shock.

The WHO considers that competent, motivated human resources as an essential pillar for providing quality maternal and newborn healthcare [54]. Our findings, similar to other studies, suggest that the COVID-19 was accompanied by a negative impact on the quality of maternal and newborn care, including delays in care provision due to PPE donning or lack of testing availability for mothers, spending less time with patients out of fear of infection because providers feel unprotected, and a reduced ability to provide emotional support to women because of the strict prevention measures [55–57]. Enabling safe working environments, including adequate infrastructure and resources, is essential to empower healthcare providers to play this indispensable role in caring for women and newborns, during a pandemic and beyond.

Our analysis provides new insights to the effect of the pandemic specifically for maternal and newborn healthcare providers. We found that almost one-third of our participants experienced a decrease in their income in months preceding the survey, with higher proportions in low-income countries (up to 70%). In these settings, limited attention has been paid to the adoption of policies aiming at mitigating the pandemic-related economic impact among health providers–inevitably related to reduced demands for healthcare and additional costs generated by personal purchases of PPE, for instance [58–60]. Such situation is known to decrease the availability and motivation of health providers in working in emergency situation, with ultimate impact on the health system, and especially the provision of maternal and newborn care during crises [61, 62].

To our knowledge, this is the first study to describe the challenges experienced by maternal and newborn healthcare providers globally and during the period beyond the early emergency response to the COVID-19 pandemic. This uniquely enabled the assessment of health systems' and actors' responsiveness in addressing challenges to the provision of essential healthcare services, which providers from many countries faced during the first period of the COVID-19 pandemic. In addition, the mixed-methods design of the study permits the use of open-text responses for better understanding quantitative data and the linkages between the providers' work-related experiences, and the various dimensions resulting from the COVID-19 pandemic. This also allowed us to conduct data quality checks by cross-verifying answer to close- and open- ended questions for consistency. Another strength of the study resides in its global scope, with respondents from 77 countries across the world. This allowed the research team to assess differences in the COVID-19 pandemic impacts across country income group. However, the interpretation of the findings could be more precise if contextualised according to the

country-level COVID-19 situation and restriction measures, which vary between and within country income groups. Future country-specific studies are recommended to explore restriction measures' impact on maternal and newborn healthcare providers.

Other limitations of our study should be noted. This global survey is a snapshot taken at one point in time. It was conducted five months after the WHO pandemic declaration and after many countries had already completed the first wave of the pandemic, and/or implemented strict infection prevention and control measures at the national level. Given the highly dynamic and changing prospects of the pandemic, including the rapid development and prioritised accessibility to health workers for COVID-19 vaccines since our data collection, variations in perceived risk of the disease, and therefore its, physical, psychological and economic impact on health providers, may exist over time. The round-two data used in this analysis benefited from questionnaire revisions after round one, however given the pandemic circumstances necessitating rapid data collection, the questionnaire was not formally validated in all settings and languages. Our survey, as many online surveys, may be associated with sampling biases whereby response rates tend to be higher around the professional networks or place of the researchers. This may explain in part why our sample is unequally distributed with a smaller number from a wide range of low-income countries. We particularly note a large proportion of responses from Kazakhstan (n = 566; 48%). To address this overrepresentation from one country, we conducted a sensitivity analysis and descriptively compared the results of the quantitative data between the full sample and a sub-sample excluding respondents from Kazakhstan (S2 Appendix). We found that in general respondents' characteristics are similar between the two samples. Some exceptions include differences in the distribution by cadre and health facility.

Our results add to the understanding of inequitable implications of the COVID-19 pandemic on global healthcare providers' well-being and motivation for the provision of quality maternal and newborn health care. The study is of particular importance as understanding the COVID-19 related impacts on healthcare providers is essential to building a responsive and stronger health system, given the central role that healthcare providers play in its functioning. However, their direct role in the provision of care to patients infected with diseases of high risk (COVID-19, Ebola, etc.) is often denied. Likewise, their potential exposure to these infectious diseases while managing undiagnosed maternity cases is overlooked. The neglect of this health workforce group could be mirrored by the variety of challenges that they continue to face during the COVID-19 pandemic including scarcity of health resources, inadequate infrastructure and working conditions which trickle down to affect the availability and quality of care provided to women, newborn and families. On the other hand, adopting strict COVID-19 prevention measures contributes to deteriorating quality of care during the pandemic and constitutes an important source of providers' psychological disorders. This means that decision-makers across the globe need to prioritise safe-guarding maternal and newborn health and wellbeing, by ensuring health providers in all income settings have both adequate PPE, mental health support, and economic hardship protection during global health crises; acknowledging their role as essential health providers, and therefore working towards limiting their shortages and rotation during health crises; putting mechanisms in place that exempt them from collateral effects of health crises measures (e.g. access to workplace during lockdown and curfew, etc.); ensuring their access to training and updated guidelines consistent with current scientific evidence. Finally, to understand and address the COVID-19 pandemic and its related impacts on healthcare providers in low-income countries, further research, especially qualitative studies, are needed.

## Conclusion

This study describes in detail the experiences of maternal and newborn healthcare providers after the first period of the COVID-19 pandemic and the interconnected and dynamic nature of its physical, psychological and economic impacts. Our data revealed that continued shortages of COVID-19 related health supplies along with negative effects on staff availability, wellbeing and motivation had a strong potential to affect the provision and quality of maternal and newborn health care globally. Healthcare providers are exposed to important physical and emotional risks in their workplace, which was in part due to aggressive behaviours from patients and/or their families. Finally, the pandemic substantially impacted health providers' income, with disproportionate effect on those living in low-income countries. This study calls on global and national policymakers for a particular attention to the implementation of interventions aiming at interrupting the acute phase of the COVID-19 pandemic in countries with limited resources.

## Supporting information

**S1 Appendix. Survey questions included in the analysis.**
(DOCX)

**S2 Appendix. Sensitivity analysis without responses from Kazakhstan.**
(DOCX)

**S1 Table. List of respondents' countries.**
(DOCX)

## Acknowledgments

We acknowledge the valuable responses of maternal and newborn healthcare providers who contributed their time to respond to the survey during the second round, despite ongoing difficult circumstances and high workload. We thank all study collaborators and colleagues who supported in questionnaire development, translation and played a key role in distributing the invitation for this survey. We also acknowledge the Institutional Review Committee at the Institute of Tropical Medicine for providing helpful suggestions on this study protocol, and for the expedited review of this study.

## Author Contributions

**Conceptualization:** Aline Semaan, Louise-Tina Day, Thérèse Delvaux, Alexandre Delamou, Lenka Benova.

**Data curation:** Delphin Kolié, Aline Semaan.

**Formal analysis:** Delphin Kolié, Aline Semaan.

**Funding acquisition:** Lenka Benova.

**Investigation:** Lenka Benova.

**Supervision:** Lenka Benova.

**Validation:** Lenka Benova.

**Writing – original draft:** Delphin Kolié, Aline Semaan.

**Writing – review & editing:** Louise-Tina Day, Thérèse Delvaux, Alexandre Delamou, Lenka Benova.

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
