## [Decision Letter · Decision Letter 0]

21 Jun 2022

PGPH-D-21-00830

Maternal and newborn healthcare providers’ work-related experiences during the COVID-19 pandemic, and their physical, psychological, and economic impacts: findings from a global online survey

Dear Authors,

Thank you for submitting your manuscript to PLOS Global Public Health. After careful consideration, we feel that it has merit but does not fully meet PLOS Global Public Health’s publication criteria as it currently stands. It requires minor revisions. Therefore, we invite you to submit a revised version of the manuscript that addresses the points raised during the review process.

Please submit your revised manuscript by July 31, 2022.  If you will need more time than this to complete your revisions, please reply to this message or contact the journal office at globalpubhealth@plos.org. Please include the following items when submitting your revised manuscript:

We look forward to receiving your revised manuscript.

Kind regards,

Shela Hirani, PhD, IBCLC, RN

Academic Editor

Journal Requirements:

1. In the Funding Information you indicated that no funding was received. Please revise the Funding Information field to reflect funding received.

Please ensure that the funders and grant numbers match between the Financial Disclosure field and the Funding Information tab in your submission form. Note that the funders must be provided in the same order in both places as well.

2. Please update your online Competing Interests statement. If you have no competing interests to declare, please state: “The authors have declared that no competing interests exist.”

Additional Editor Comments (if provided):

Reviewers' comments:

Reviewer's Responses to Questions

**Comments to the Author**

1. Does this manuscript meet PLOS Global Public Health’s publication criteria? Is the manuscript technically sound, and do the data support the conclusions? The manuscript must describe methodologically and ethically rigorous research with conclusions that are appropriately drawn based on the data presented.

Reviewer #1: Yes

Reviewer #2: Yes

Reviewer #3: Yes

2. Has the statistical analysis been performed appropriately and rigorously?

Reviewer #1: Yes

Reviewer #2: Yes

Reviewer #3: I don't know

3. Have the authors made all data underlying the findings in their manuscript fully available (please refer to the Data Availability Statement at the start of the manuscript PDF file)?

Reviewer #1: No

Reviewer #2: Yes

Reviewer #3: No

4. Is the manuscript presented in an intelligible fashion and written in standard English?

Reviewer #1: Yes

Reviewer #2: Yes

Reviewer #3: Yes

5. Review Comments to the Author

Reviewer #1: Overall, this is a well-written manuscript. The authors have systematically examined the topic of investigation. Comparison of the results across countries of different income levels is useful in mapping the areas with greater impact. With such a high number of sample size, it would have been desirable to perform some advanced statistical analysis to identify underlying associations.

There are a few specific comments as mentioned below.

1. None of the findings from the qualitative data are included in abstract. Since the study claims to be a mixed-method study, the abstract needs to include both quantitative and qualitative findings.

2. The term ‘frontline healthcare providers’ used in concluding sentences of abstract (Line 58) is misleading. This term also encompasses community level health workers in many countries and this study does not include all those type of health workers.

3. Was the tool used piloted and validated? Were the translated versions in different languages validated? This should be part of methods and/or limitations section.

Reviewer #2: General Observation- It needs to be concise and crisp.

Methods-Following questions/concerns needs to be addressed under methodology section-

•What measures have been taken for ensuring data collection quality?

•Description of 551 qualitative responses can be given separately, like it was given for quantitative responses. How many responses received from high-,middle- and low-income countries.

•How open text responses were analyzed, manually or through software

Result

•What were the new themes emerged under open text response (Classification by type of respondents, A matrix can be provided)

Discussion

•Currently dominated by findings of low-income countries, how high- and middle-income countries were affected that can also be discussed.

•Context of covid 19 pandemic during study time in high, low- and middle-income countries can also be specified, whether it was complete lock down or restrictions were imposed.

•How the condition of different level of health providers during the pandemic impacted the quality of newborn care can be focused under discussion section.

Reviewer #3: in my point of view, this manuscript will meet PLOS Global Public Health’s publication criteria with some improvement.

from a statistical point of view, it would be better to show to someone who is an expert in the field.

6. PLOS authors have the option to publish the peer review history of their article (what does this mean?). If published, this will include your full peer review and any attached files.

**Do you want your identity to be public for this peer review?** For information about this choice, including consent withdrawal, please see our Privacy Policy.

Reviewer #1: No

Reviewer #2: No

Reviewer #3: **Yes**

---

## [Editor Report · Decision Letter 1]

7 Jul 2022

Maternal and newborn healthcare providers’ work-related experiences during the COVID-19 pandemic, and their physical, psychological, and economic impacts: findings from a global online survey

PGPH-D-21-00830R1

Dear Authors,

We are pleased to inform you that your manuscript 'Maternal and newborn healthcare providers’ work-related experiences during the COVID-19 pandemic, and their physical, psychological, and economic impacts: findings from a global online survey' has been provisionally accepted for publication in PLOS Global Public Health.

Best regards,

Shela Hirani, PhD, IBCLC, RN

Academic Editor